# A Cross-Sectional Description of Parental Perceptions and Practices Related to Risky Play and Independent Mobility in Children: The New Zealand State of Play Survey

**DOI:** 10.3390/ijerph16020262

**Published:** 2019-01-17

**Authors:** Charlotte Jelleyman, Julia McPhee, Mariana Brussoni, Anita Bundy, Scott Duncan

**Affiliations:** 1Human Potential Centre, School of Sport and Recreation, Auckland University of Technology, Auckland 0632, New Zealand; julia.mcphee@aut.ac.nz (J.M.); scott.duncan@aut.ac.nz (S.D.); 2Department of Pediatrics, School of Population and Public Health, University of British Columbia, British Columbia Children’s’ Hospital Research Institute, British Columbia Injury Research and Prevention Unit, Vancouver, BC V6H 3V4, Canada; mbrussoni@bcchr.ubc.ca; 3Professor and Department Head, Occupational Therapy, Colorado State University, Fort Collins, CO 80523, USA; anita.bundy@colostate.edu; 4Professor of Occupational Therapy, Faculty of Health Sciences, University of Sydney, Camperdown, NSW 2006, Australia

**Keywords:** free play, real play, physical activity, risk tolerance, injury prevention, outdoor play, stranger danger

## Abstract

The potential for risky play and independent mobility to increase children’s physical activity, and enhance cognitive development and emotional wellbeing has been recognised for some time. The aim of this study was to describe the attitudes of New Zealand parents towards such risky play practices and independent mobility, the barriers preventing them from allowing their children to participate, and how often their children engaged in risky play activities. An online survey comprised mostly of validated scales and standardised questions was completed by a nationally representative sample of 2003 parents. We found that parents had neutral feelings about the risk of injury to their child through play, rather they were concerned about road safety and “stranger danger”. There was strong agreement that there are multiple benefits to be gained from exposure to risk and challenge, and that health and safety rules are too strict. However, 73% of respondents stated that their 5–12 year old child seldom or never engaged in four or more risky activities, and only 14.3% engaged in four or more *often* or *always*. While parents agree that their child is likely to benefit from risky play, they do not have the confidence to allow their children to engage in such activities. Future research should address barriers and fears when implementing strategies to facilitate risky play.

## 1. Introduction

Physical activity is essential for both physiological and psychological healthy development in children [1,2]. Reports suggest that, on average, New Zealand children [3] and adolescents [4] currently meet the global recommendations for moderate-vigorous physical activity (MVPA) of 60 minutes per day, five days a week [5]. However, results from the New Zealand Report Card on Physical Activity for Children and Youth indicate that, in parallel with global trends [6], these levels are decreasing [7]. Given that physical activity habits in early childhood are likely to predict those in adulthood [8,9], it is pertinent to promote strategies that encourage and increase physical activity in children.

While the benefits of play have been documented for some time [10,11], the value of play has only relatively recently been formally recognised as being vital for a child’s healthy emotional, social and intellectual development [12]. In fact, given the strong evidence that all types of play contribute to a child’s self-esteem, behaviour regulation, emotional expression, cognitive and motor skills, resilience and mental well-being, [12,13] play is identified as a child’s right in the United Nations Convention on the rights of the child [14]. Risky, outdoor play—play that is challenging, thrilling, and involves some physical risk [15]—has been cited as a potential allure to engage children in physical activity, potentially increasing MVPA [16,17] and improving psychological well-being [13], [18]. Emerging literature has identified six categories of risky play: play with great heights, play with high speed, play with dangerous tools (e.g., hammer, saw), play near dangerous elements (e.g., water, fire), rough-and-tumble play (e.g., play fighting) and play where children can disappear or get lost (in older children, this is sometimes referred to in the literature as independent mobility) [19].

Given the complex, multi-factorial nature of children’s development, and the inherent difficulty in monitoring play practices and their effects, it has been difficult to discern whether there is a causal link between engagement in risky play and long-term physical and mental well-being. Despite this, most studies indicate that substantial benefits [10,11,20] are likely to be gained, with few showing, at least in the short term, negative consequences of minor injury, verbal disputes or physical conflict resolution [15,21,22]. Moreover, it could be argued that the importance of avoiding acute physical harm (often minor) is prioritised over the prevention of long-term psychological distress [23]. There is an argument that autonomy and risk management skills are a requirement for healthy development. For example, there is an association between reduced opportunity for all aspects of risky play and increased incidence of behavioural and psychological issues in children and young adults. Furthermore, profound positive effects of free play interventions have been observed in children with ADHD [24,25]; however, these conclusions have not been demonstrated empirically.

Despite these potential benefits, opportunities to engage in risky play behaviours have declined over the last three decades [26,27,28]. In many developed nations, participation in what is considered risky play has shifted to structured, supervised, and/or indoor activities [29,30,31]. This appears to be, in part, the consequence of increasing concerns about child safety and accountability issues [32,33,34]. Such concerns have also led to increased restrictions of children’s independent mobility (roaming the neighbourhood unsupervised), active transport to/from school, and leisure activities with friends [27,31,35]. The decrease in independent mobility is a concern because not only is this a convenient way to accumulate MVPA [36], but also by learning to navigate the neighbourhood, skills such as problem solving and decision making, for example when crossing roads, are developed [37]. Additionally, when children interact with peers away from adults they are compelled to resolve their own conflicts thus fostering social competence [38]. Notably, statistics indicate that incidents of ‘stranger danger’ have not changed over time [39,40,41,42], and current advice is actually to dismiss this mantra as the overwhelming majority of abductions, which in any case are rare, are made by adults whom the child knows [43] and not strangers. Furthermore, while traffic has increased, child road fatality in New Zealand has actually dropped over the last 30 years [44].

Numerous studies indicate that caregivers, teachers, and some policy-makers acknowledge the benefits of risky play [45,46,47] and independent mobility [48,49], yet whether or not children are allowed to engage in such activities seems to be dependent on the child’s age [34], cultural norms [50,51,52], and local policy [46,53]. For instance, as children grow older, the number of risky activities in which they are permitted to participate increases [34,54]. In addition, teacher and parent willingness to encourage children’s participation in risky play appears to align with the strictness of the safety standards of regulated institutions, such as nurseries and schools [55].

In New Zealand, there has been a notable decline in children’s outdoor play and active travel [4,28], but we still have no descriptive information about the perceptions and practices related to risky play. The aim of this study was to describe how a large sample of New Zealand parents perceive risky play behaviours in children, including children’s independent mobility, as well as the level of risky play permitted by parents. We hypothesised that risky play and independent mobility would be perceived as beneficial for children, but that this perception would not necessarily be reflected in parenting practice.

## 2. Materials and Methods 

This study reports a descriptive analysis of the results of the New Zealand State of Play Survey, a cross-sectionally implemented questionnaire. Ethical approval was obtained by Auckland University of Technology Ethics Committee (AUTEC; 15/227 The way we use to play: Promoting free play in New Zealand children. Approved 6 July 2015). This study has been reported according to the STROBE guidelines [56]. The checklist can be found in the Appendix A.

### 2.1. Participants and Setting

The Smile City Research Panel (Kantar TNS Global, Auckland, New Zealand branch), a database of individuals residing in New Zealand, was used to identify active members listed as having at least one dependent child younger than 18 years. Recruitment and data collection were carried out by an independent market research company (TNS Global) between 19–31 August 2015. 

### 2.2. Eligibility and Inclusion Criteria

Participants were excluded from the analyses if (1) they stated in the first screening question that they were not a parent or caregiver over the age of 18 with at least one dependent child, or (2) they did not complete the survey. Recruitment and inclusion flow are depicted in Figure 1. Using TNS Global in-house methodology, 13,400 eligible panel members were randomly selected to take part. In total, 2546 (19%) questionnaires were returned, with 2003 (14.9%) meeting the minimum criteria for analysis. A total of 1573 (11.7%) respondents had children aged 5–12 who were referred to the risky play questionnaires, and 1366 respondents had children aged 9–10 about whom independent mobility questions could be answered.

### 2.3. Procedures

Email invitations were sent to potential participants who were asked to fill out an online survey via the Dimensions (IBM) platform using a point-and-click interface, visually and functionally similar to a paper-based questionnaire. Some questions referred to children of a certain age; parents without children in this age-range were automatically skipped through these sections of the survey. 

### 2.4. State of Play Survey

The State of Play Survey, funded by Persil NZ, comprised several questionnaires designed to determine parents’ perceptions and practices of risky play and independent mobility (please see Appendix A). Risky play was defined as play meeting the characteristics described by Sandseter et al. [56] (Table 1). After consultation with play experts, two further categories were included: play with loose objects (e.g., sticks, timber, tyres, tarpaulins) and ‘messy’ play [57].

#### 2.4.1. Tolerance for Risk in Play Scale

The Tolerance for Risk in Play Scale (TRiPS) [58] was developed to ascertain parents’ tolerance for children to experience risk during play. The TRiPS is a 32-item questionnaire that asks respondents whether or not they do allow their children to engage in activities pertaining to each category in Table 1. A few questions also refer to supervision and child injury risk, everyday risk propensity and parental over protection. The survey asks parents to consider their responses to various risky scenarios with reference to their eldest child aged between 5–12 years (parents of children aged 13–18 years were automatically skipped through this part of the survey). Sample items include: “Would you let the child go head first down a slippery dip?” and “Would you let the child walk on slippery rocks close to water?” Parents answer questions either “yes” or “no”. “No” answers are scored 0 and yes answers are scored from 1–12 weighted according to their acceptability or difficulty to endorse based on a Rasch analysis of a validation study [58], with the least acceptable statements scoring higher than the most acceptable statements. It should be noted that while higher scores indicate greater risk tolerance, this scale is not strictly continuous as not all values between 0 and 184 are possible totals.

#### 2.4.2. Risk Engagement and Protection Survey

The Risk Engagement and Protection Survey (REPS) [59] was designed to help understand the views and attitudes of parents towards protecting children from injury and allowing them to engage in risks. To date, it has only been validated for fathers [58], although is currently being assessed on mothers. Therefore, given that there is currently no equivalent scale for female caregivers, both male and female respondents were asked to answer this section of the survey. The survey contains 12 items: six relating to protection from injury and six relating to risk engagement. Participants are asked to rate how much they agree or disagree with each statement using a seven-point Likert scale anchored with “very strongly agree” and “very strongly disagree”.

#### 2.4.3. Perception of Positive Potentiality of Outdoor Autonomy for Children Scale

An adapted version of the Perception of Positive Potentiality of Outdoor Autonomy for Children (PPOAC) scale [60] was included to establish the extent to which parents believed that roaming the neighbourhood was positive or negative for 9–10 year-olds as this is the age group in which the questionnaire has been validated in. Answers are given on a four-point Likert scale ranging from “strongly agree” to “strongly disagree”. There are nine items meaning that possible scores range from nine to 36. Sample items include: “Become more responsible” and “Encounter ill-intentioned adults”. For analysis, negative statements were reverse coded so that ‘positive’ attitudes resulted in positive score increases.

#### 2.4.4. Extraneous Barriers to Risky Play

Participants were asked to state to what extent they agreed with statements regarding barriers to allowing children to engage in risky play. Responses were rated on a five-point Likert scale from “strongly disagree” to “strongly agree”. Examples of these statements include: “Finding ways to get children active is expensive these days” and “There are too many unnecessary safety rules in New Zealand schools” (see Appendix A; Q6).

Two questions relating to playing in the rain were included in the survey. The first pertaining to the frequency with which a child plays in the rain was rated on a five-point scale from “never” to “always”. The second suggested eight reasons a child might not be allowed to play in the rain and respondents were asked to indicate which might influence their decision. Items covered concerns about sickness and injury, appropriate clothing and dislike for being in the rain. There was also an option for “other”.

#### 2.4.5. Participation in Risky Play Activities

To determine whether perceptions of risky play changed depending on a child’s age, participants were asked at which age (if at all) a child should first be allowed to engage in each category of risky play. Example items include: “Climb trees at home or in their local park or recreation areas” or “Use adult tools (e.g., hammer, saws, drills) at home”. The purpose of asking parents to state the appropriate age for each play/mobility category is that it enabled a relatively graded response facilitating comparisons of perceptions among categories. They were then asked to indicate on a five-point Likert scale, how often (from never through seldom, sometimes and often to always) their eldest child aged 5–12 engages in a selection of activities with varying degrees of supervision. Six of the activities were examples of each of the risky play categories described by Sandseter [19] for example: “How often does your child use adult tools?” (see Appendix A; Q13). Parents were also asked how often their child rides non-motorised vehicles in three different scenarios: under the supervision of adults, with friends or alone. Two items were specifically included to gain an understanding of the child’s independent mobility in terms of roaming the neighbourhood.

#### 2.4.6. Active Transportation and Independent Mobility

Independent mobility is a sub-category of risky play as children considered “streetwise” appear to be trusted by their parents to make responsible decisions if faced with risky situations [61]. Respondents were asked questions about how their eldest child between age 5–12 normally travels to and from school, who they travel with and, for children aged 9–10, under what circumstances, if any, they are allowed to travel to other destinations unsupervised by adults. Questions were adapted from those used in a previous report comparing the independent mobility of children in England and Germany between 1971–2010 [52]. Respondents were asked to answer “yes”, “no”, or “N/A” to six items such as “Is your child usually allowed to go out alone after dark?” and “Is your child usually allowed to travel on local buses alone (other than a school bus)?” This questionnaire has recently been validated [62].

### 2.5. Statistical Analysis

Cross-sectional, descriptive analyses were carried out using R V3.5.0 (R Core Team (2013). Vienna, Austria). Survey response distributions were checked for normality using visual interpretation of histograms or, if necessary, Shapiro-Wilk tests. Where the assumption of normality was violated, medians and interquartile ranges are reported. The number of people who rated each level for each item was counted for all questionnaires. In addition, the following analyses were conducted. Quartiles were calculated for TRiPS scores to create four categories of risk tolerance: risk averse, somewhat risk averse, somewhat risk tolerant, risk tolerant. The proportion of “yes” answers of each weighting was then calculated for each category. For the REPS, the number of items rated high (>6) or low (<2) out of seven by each participant was counted. Ratings for each answer in the PPOAC scale were summed to produce a total score out of 36. The number and proportion of respondents agreeing with each barrier to playing in the rain was calculated. Furthermore, the number of activities participated in at all (independent mobility), or frequently (>5) or infrequently (<2; risky play) was counted. All figures were created using R package ggplot2 [63].

## 3. Results

### 3.1. Participants

Participant characteristics are summarised in Table 2. Compared to the New Zealand population, the sample was representative in terms of number of children, ethnicity, household income and dwelling location [64].

### 3.2. Survey Questionnaires

Summary statistics for survey scores are presented in Table 3. Of the final sample, 1573 (78.5%) completed the validated questionnaires in full. Independent mobility information was available for 1366 participants.

#### 3.2.1. Tolerance for Risk in Play

Median (LQ–UQ) score on the TRiPS scale was 95 (61–122) out of 184. A median of 17 (6–23) out of 30 statements were answered “yes”, indicating that, on average, parents would allow their child to engage in 55% of the included risky play activities. TRiPS total scores associated with the risk category quartiles are presented in Figure 2a and Table 3. As expected, as item weighting increased, the number of respondents agreeing with the statement decreased Figure 2b). Respondents categorised as risk averse (score ≥ 41) tended to answer more, lower-scoring questions rather than a few higher-scoring questions (Figure 2b).

#### 3.2.2. Risk Engagement and Protection

Distribution of answers for injury prevention and risk engagement are presented in Figure 3a,b, respectively. Most parents disagreed to some extent that they were concerned about hazards in the home. Almost exactly half (50.2%) of parents agreed or strongly agreed that good supervision means knowing what their child is doing at all times. The remaining four items relating to injury prevention had a median rating of “neither agree nor disagree”. Of those who had a complete set of survey answers, only 3.3% very strongly disagreed or strongly disagreed with most (four or more) items that state that injuries should and could be avoided by intensive supervision. Similarly, only 3.4% very strongly agreed or strongly agreed with this sentiment. Most respondents (87.1%) marked most statements between three to five out of seven suggesting moderate views on avoiding injury by managing play environments.

All risk engagement statements had a median rating of “agree” demonstrating that overall, risk engagement was viewed positively. For instance, whereas only 0.7% of respondents very strongly or strongly disagreed with statements proposing that risk and challenge were beneficial, 28.6% marked most statements “strongly agree”, or “very strongly agree”. This indicates that perceptions towards risk engagement are strong and positive for a significant proportion of individuals.

#### 3.2.3. Perception of Positive Potentiality of Outdoor Autonomy for Children (PPOAC)

The number of parents responding with each rating to the items in the PPOAC scale is presented in Figure 4. The median score in the PPOAC survey was 24 indicating that most parents perceived outdoor autonomy as “likely” to be positive. Overall, statements regarding the potential for outdoor autonomy to encourage childhood development were rated as “likely”. However, while 70.8% of parents thought it was unlikely that a child would feel disorientated, 73.2%, 56.3% and 59.9% believed that road accidents, stranger danger and seeing scary things, respectively, was likely or very likely. Interestingly, though, 74.6% of parents thought it was likely or very likely that their child would be able to find someone to help in case of trouble.

#### 3.2.4. Extraneous Barriers to Risky Play

All 2003 participants had a complete set of responses for questions relating to perceptions of health and safety. Ratings are captured in Figure 5. More than half the respondents either “agreed” or “strongly agreed” that there are too many health and safety rules in schools (55.9%) and regarding children’s play in general (60.4%). More than three quarters (78.6%) of participants “agreed” or “strongly agreed” that regular exposure to risk develops risk management skills and 74.9% “agreed” or “strongly agreed” that exposure to actual risk enhances children’s development. Furthermore, only 24.0% “agreed” or “strongly agreed” that relaxing rules would increase the risk of serious injury.

Nearly 14% of parents said that they never let their children play in the rain (see Figure 6a). There did not appear to be strong opposition to allowing children to play in the rain, however, with 57.8% of parents always, often or sometimes allowing it. Of the provided options, the most commonly selected barriers were that rain would cause children to get sick (N = 811) and make them too cold (N = 779). Half this number of parents (N = 405) believed that their child(ren) did not like playing in the rain. Getting messy (N = 179) and not having appropriate clothing (N = 134 child; N = 52) did not emerge as primary concerns (see Figure 6b).

#### 3.2.5. Participation in Risky Play Categories

Results showed that the older children get, the more likely parents are to allow them to participate in risky play. Most children (73.4%) were allowed to engage in messy play by age three; climb trees (67.4%), engage in rough and tumble play (60.9%), and ride non-motorised vehicles (54.1%) at age five, and use adult tools (61.9%) at home by age nine. Most children were not allowed to roam the neighbourhood unsupervised by adults (but with friends; 67.6%) until age 13, or alone (64.9%) until age 15.

Fewer than half of parents reported that their children participate in at least one of the six activities (see Figure 7) “often” or “always”. Moreover, it was more common for parents to report that their child “seldom” or “never” participates in at least five of the eight activities than for them to engage in them “often” or “always” (27.2% vs. 14.3%). However, parents were more likely to allow their children to engage in the risky play activities: climbing trees, playing with tools and loose parts, engaging in rough and tumble, riding bikes and messy play, than roaming the neighbourhood alone. Most parents allow their children to participate in most of the risky play activities at least sometimes or often or always. This was not the case for independent mobility, however, with only 14.5% of children allowed to roam the neighbourhood with friends sometimes, often or always. This proportion dropped to 6.7% if the child was alone.

#### 3.2.6. Active Transportation and Independent Mobility

Data were available for 1366 participants who had children aged nine or ten. The mode by which and with whom children travel to school is presented in Table 4. Only 364 (26.7%) children traveled without an adult, and 118 of these (32.4%) used the bus where there is some degree of supervision from the driver. Of the 725 (53.1%) children who traveled actively, 320 (44.1%) traveled either alone (157; 21.7%) or with siblings or friends (163; 22.5%).

Fewer than half of respondents indicated that their child was allowed to roam the neighbourhood, or cross main roads alone. Fewer than one fifth were allowed to cycle on main roads and fewer than one eighth were allowed to use the bus for travel other than to school. Just 4% were allowed out alone after dark.

## 4. Discussion

This study represents the first observation of a large, nationally representative sample of New Zealand parents’ perceptions of risky play and the frequency with which their children engage in risky play activities. Using recently validated questionnaires, we report a rich set of results covering multiple aspects of risky play and independent mobility perceptions and practice. We found that, overall, parents perceived exposure to risk as beneficial for children’s development. However, participation in risky play activities was limited with most children only ‘sometimes’ taking part in these types of activities. As expected, as children grew older, parents were more willing to allow them to engage in risky play. With regard to health and safety, parents thought that there were too many rules and that relaxing them was unlikely to increase injuries. Interestingly, while parents were neutral about whether less supervision would increase the risk of injuries, they believed that good supervision meant knowing what their child is doing at all times. Furthermore, the dangers of unsupervised roaming were rated as likely and children aged 5–12 years were seldom or never allowed to roam the neighbourhood without adults. Fewer than half of respondents afforded their children independent mobility to roam the neighbourhood on foot, with this number decreasing exponentially when asked about using bicycles, the bus and going out after dark.

To the best of our knowledge, this is the first study to assess New Zealand parents’ attitudes towards risky play quantitatively using validated questionnaires. The finding that there was a dichotomy between the perceived benefits of risky play and actual participation is consistent with views of parents from Australia [23,65] who revealed in semi-structured interviews that they find it difficult to balance the importance of taking risks with managing safety. The general consensus that risk is beneficial, injury unlikely, but supervision necessary, demonstrates a lack of confidence in parents’ facilitation of risky activities. Indeed, fear of being judged to be a bad parent has been reported to be one motivator of intensive supervision [66]. We found that parents did not regard the cost of activities as a barrier. Instead, perhaps paradoxically, whereas health and safety rules were considered too strict, parents were also strongly concerned about the risk of road traffic accidents and stranger danger, suggesting that enforced safety measures do not address parent’s concerns. This is supported by qualitative studies from New Zealand where increased traffic volume is identified as one barrier to allowing children to play unsupervised or use non-motorised vehicles outside, particularly in more affluent areas [28,67]. Fear of encounters with ill-intentioned adults have also been echoed in a number of studies and may partly stem from a reduced sense of community with family units believing themselves to be more isolated from their neighbours than they used to be. Parents report that when they were young, everyone looked out for each other’s children, but that this type of neighbourhood surveillance does not exist as much anymore [28,68]. This was also found in a study from Australia, with perceived “social control” predicting independent mobility [69].

Parents’ views that there are too many health and safety rules applied to children’s play contrasts with those of early childhood educators who believe that one of the strongest drivers for restrictions on play equipment is pressure from parents [23,70]. Many examples of parents blaming teachers for minor injuries their children have sustained have been reported [45] with isolated cases appearing to dictate institutional policy [71]. Sentiments expressed in previous studies suggest that parental identity, such as the role of protector [72] and social judgement [23,34,66] are important motivators in parents’ decisions regarding their children’s freedom. Such conflicting views highlight societal pressures to be viewed as though one is caring appropriately for the children within their charge, as opposed to actually allowing what is best for the child. It is vital, therefore, that institutions of power and influence such as the media, playground inspectors and local authorities set examples and standards that consider the long-term well-being of the child rather than the acute social discomfort of the caregiver [50].

It was not possible in this study to discern the reasons why respondent parents’ children do not regularly participate in risky play. However, comments from North American [30] and Norwegian [73] parents discussing outdoor free play highlight busier school schedules (e.g., homework, music, organised sport) and increased use of technology as barriers to unstructured play. Notably, these activities were not necessarily viewed as detracting from children’s development, rather the skills gained thought of as essential for the future and preventative of troublesome behaviour [30], a sentiment also captured by a New Zealand study [28].

To date, the research pertaining to risky play practices (including outdoor, nature and unsupervised real play) has mostly been qualitative and/or has employed questionnaires developed specifically for individual studies [73,74]. Risk perception has been shown to be culturally specific and it is, therefore, difficult to compare degrees of risk tolerance. For example, parents and teachers from Norway and Canada appear to be less risk averse than those from the USA and Australia [30,51,75]. Due to its geographical topography, with the ocean and bush accessible to a large proportion of the population, in the international context New Zealand is also likely to emerge as a relatively risk tolerant society although, until now, this has been impossible to demonstrate empirically. Nonetheless, New Zealand residents still feel as though attitudes towards children’s play are restrictive. We believe this study is a valuable addition to the literature as the use of three validated questionnaires mean that the results can be compared at an international level.

While the use of validated questionnaires is a strength of this investigation, currently there are no other studies that have reported analogous results and, therefore, we have no way of comparing our findings in the wider context. In addition, using Likert scale responses, especially four- and five-point scales is restrictive when attempting to understand participant opinion. Furthermore, respondents were not always given the option to add items or activities that were not listed. Therefore, influences that they felt were important or other risky activities that their children took part in were not captured in this study, and thus there is a possibility that this report underestimates participation in risky activities. Correspondingly, while this study suggests that parents are open to the idea of risky play, it neither discerned to what extent nor provided possible solutions as to how to facilitate engagement. It would also be useful to understand what factors predict participation, and future research should attempt to identify which socio-demographic and individual characteristics influence risky play practices. It should also be noted that there are inherent limitations to online survey methodology including response rates, just 19% in this study, with the possibility that those with the strongest opinions on the topic respond.

## 5. Conclusions

In conclusion, we found that parents believe that risky play enables numerous benefits to children’s development, but many do not allow their children to take part in such activities. Given that this type of play has great potential to improve both physical and psychological health, both pertinent issues increasingly affecting each generation of children [76,77], this topic is worthy of further investigation. Future studies should thus investigate what modifiable factors predict participation in risky play, as well as investigate the influence of cultural norms and societal pressures so that the apparent, inhibited desire of parents to allow their children more freedom may be facilitated and accepted.

## Figures and Tables

**Figure 1 ijerph-16-00262-f001:**
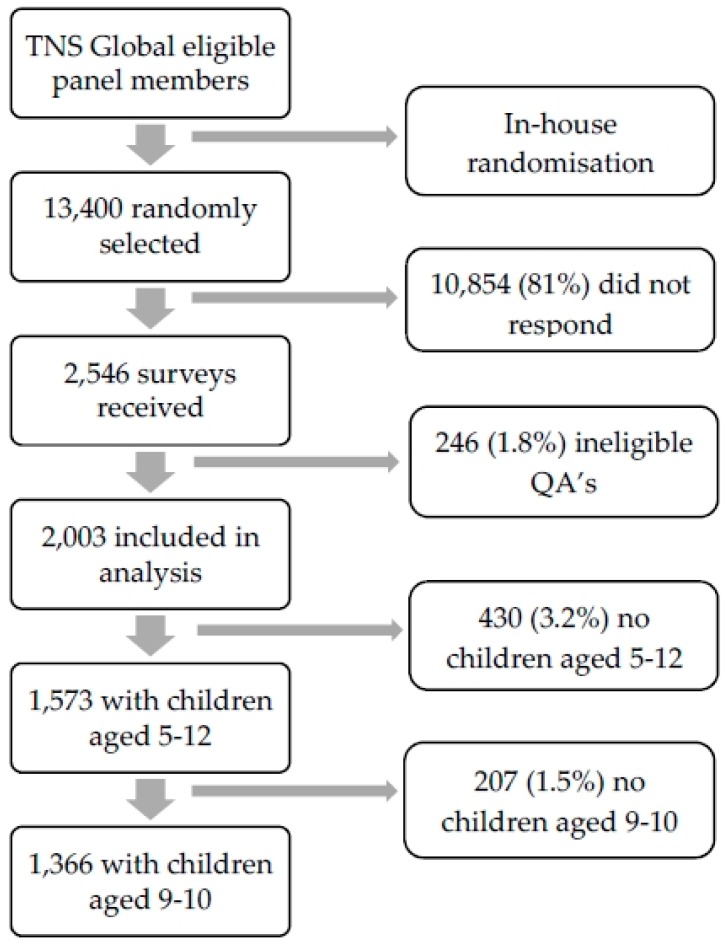
Participant recruitment and inclusion.

**Figure 2 ijerph-16-00262-f002:**
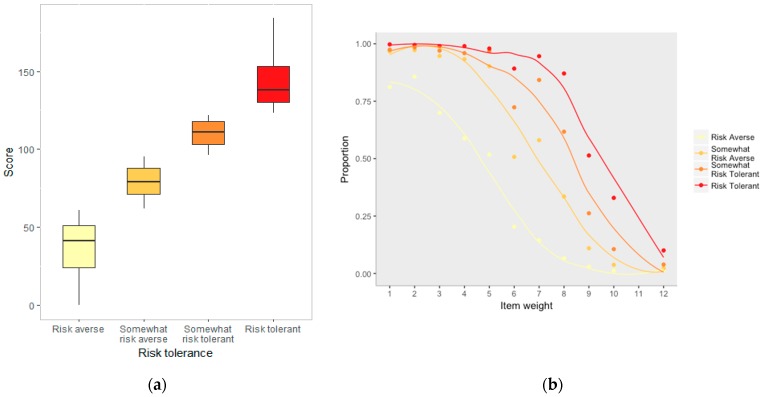
Results from the Tolerance for Risk in Play Scale (TRiPS) (**a**) Scores associated with each quartile (N = 398 in each group) and (**b**) Relationship between number of yes answers and item score for each risk category (N = 1573).

**Figure 3 ijerph-16-00262-f003:**
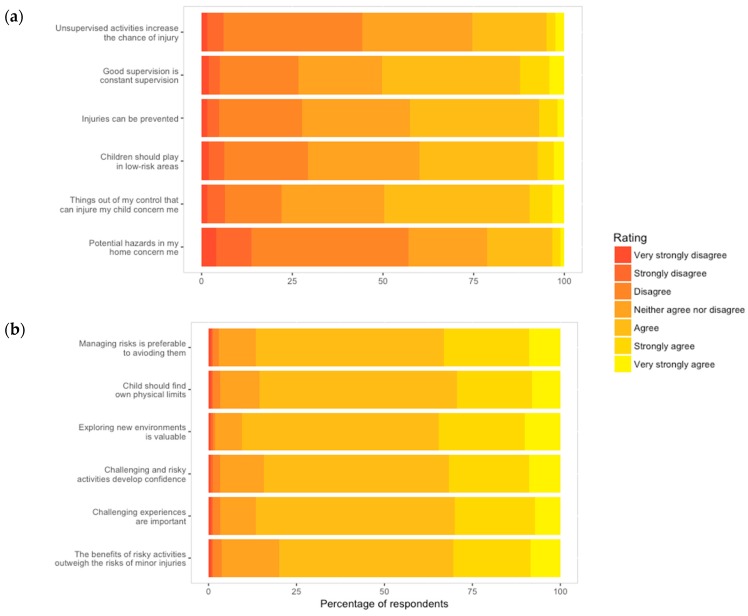
Risk Engagement and Protection Survey (REPS) responses to (**a**) injury prevention and (**b**) risk engagement questions.

**Figure 4 ijerph-16-00262-f004:**
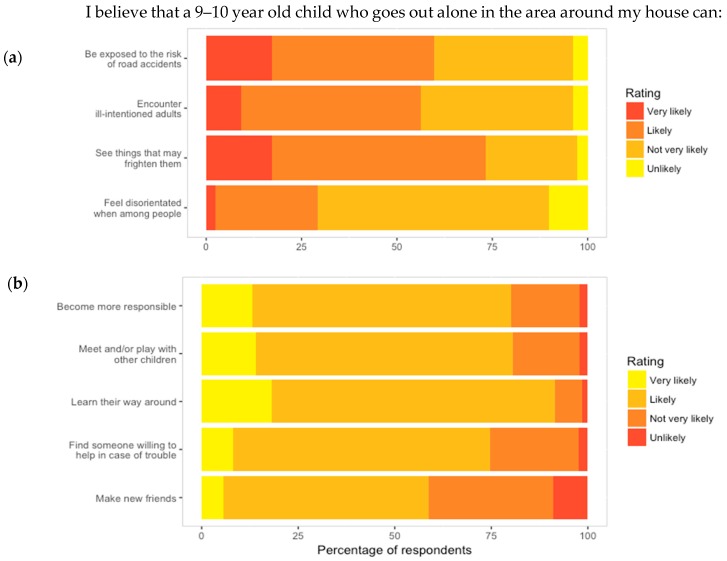
Responses to items from the Perception of Positive Potentiality of Outdoor Autonomy for Children (PPOAC) scale. (**a**) Negative statements and (**b**) positive statements.

**Figure 5 ijerph-16-00262-f005:**
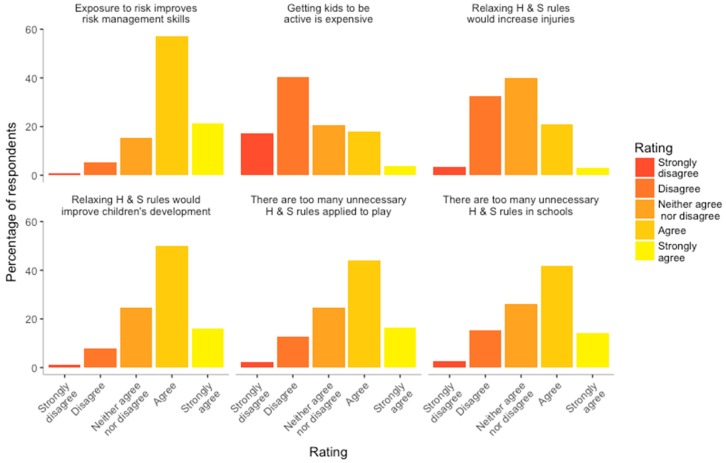
Parents attitudes to health and safety rules (n = 2003). H&S—health and safety.

**Figure 6 ijerph-16-00262-f006:**
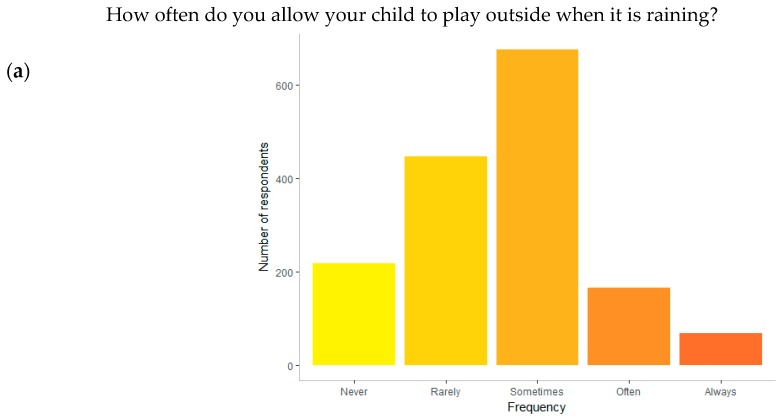
(**a**) Number of parents who allow their children to play in the rain (N = 1573) and (**b**) proportion of respondents agreeing with the suggested barriers. Respondents could mark as many options as they wished. All “yes” answers were summed (total = 3001) and the number per barrier calculated.

**Figure 7 ijerph-16-00262-f007:**
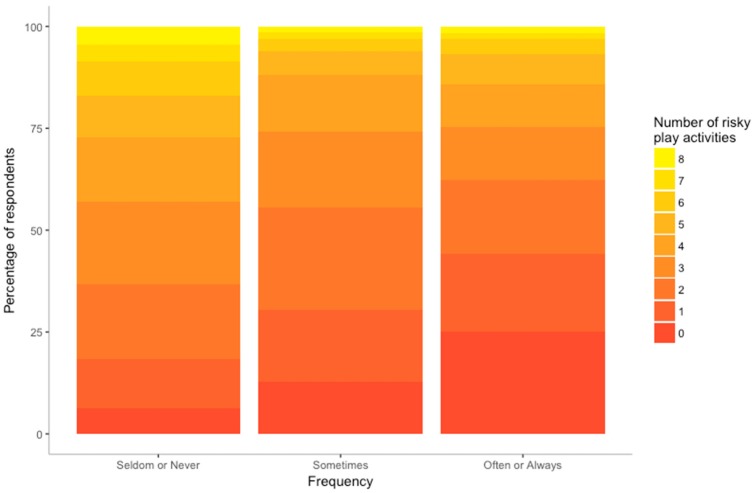
Number of risky play activities engaged in infrequently (seldom or never), sometimes and frequently (often or always) by children aged 5–12 (N = 1573). The total number of participants responding with each frequency rating was calculated across all activities and expressed as a percentage according to the number of activities that received that rating. For example, 234 respondents reported that their child seldom or never engaged in four of the eight activities.

**Table 1 ijerph-16-00262-t001:** Definitions of risky play used in the State of Play Survey.

Category	Risk	Examples
**Great heights**	Danger of injury from falling	Climbing, jumping, balancing, hanging, swinging
**High speed**	Uncontrolled speed and pace that may lead to collision	Swinging, sliding/sledging, or non-motorised vehicles
**Adult tools**	Potential for injury or wounds	Knives, saws, axes, drills, ropes
**Dangerous elements**	Risk of injury from falling into or from something	Trees, cliffs, water, fire
**Rough and tumble**	Children may harm each other	Play-fighting, wrestling, fencing with sticks
**Disappear or get lost**	Children are unsupervised, alone or lost	Roaming neighbourhood with friends or alone, exploring
**Loose parts ^†^**	Danger of injury from sharp or heavy objects. Use of dirty objects	Tyres, sticks, timber, tarpaulins
**Messy play ^†^**	Illness from unsanitary environments	Painting, play in mud, dirt, sand, water

Adapted from Sandseter et al. (2011), ^†^ Added to broaden the concept of risky play [57].

**Table 2 ijerph-16-00262-t002:** Demographic characteristics of respondents (N = 2003).

Characteristic	N (%)
Gender	
Male	629 (31.4)
Female	1374 (68.6)
Parent age (years)	
Under 30	192 (9.6)
30–39	631 (31.5)
40–49	799 (39.9)
Over 49	381 (19.0)
Number of children	
One	675 (33.7)
Two	823 (41.1)
Three	349 (17.4)
Four or more	157 (7.7)
Eldest child’s age	
0–4	53 (2.6)
5–8	299 (14.9)
9–12	512 (25.6)
13–16	540 (27.0)
16+	599 (29.9)
Ethnicity	
New Zealand European	1412 (70.5)
New Zealand Maori	234 (11.7)
Pacific Island	114 (5.7)
Asian	146 (7.3)
Annual household income	
Less than $40.000	373 (18.6)
$40.000–$100.000	855 (42.7)
More than $100.000	431 (21.5)
Location	
Large city	939 (46.9)
Small city	457 (22.8)
Town	377 (18.8)
Small town	102 (5.1)
Rural	122 (6.1)

**Table 3 ijerph-16-00262-t003:** Survey summary statistics (n = 1573).

Questionnaire (Score Range)	Median (LQ–UQ)
Tolerance of risk in play scale (0–184)	95 (61–122)
Risk averse (0–61)	41 (24–51)
Somewhat risk averse (62–95)	79 (71–88)
Somewhat risk tolerant (96–122)	111 (103–118)
Risk tolerant (123–184)	138 (130–153)
Risk engagement and perception survey	
Injury Prevention (6–42)	24 (21–27)
Risk engagement (6–42)	30 (29–34)
Perception of positive potentiality of outdoor autonomy (9–36) ^†^	25 (23–26)

^†^ Possible scores once reverse coded for negative statements. Higher scores assigned to positive answer indicating lower fear.

**Table 4 ijerph-16-00262-t004:** Children’s travel to school and independent mobility (N = 1366).

	N (%)
Mode:	
Active (e.g., walk, bike, skateboard)	725 (53.1)
Passive (e.g., car, bus)	641 (46.9)
Accompaniment ^†^:	
Alone	361 (26.4)
Friends	387 (28.3)
Siblings	509 (37.3)
Adult	1002 (73.4)
Roaming the neighbourhood:	
Destinations within walking distance	587 (43.0)
Crossing main roads	605 (44.3)
Out alone after dark	55 (4.0)
Cycling on main roads	259 (19.0)
Using buses (not for school)	181 (13.3)

^†^ Respondents could choose more than one option and therefore this total is >1366.

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
