# Peer review of "A Cross-Sectional Description of Parental Perceptions and Practices Related to Risky Play and Independent Mobility in Children: The New Zealand State of Play Survey"

_ijerph, 2019, doi:10.3390/ijerph16020262_

Reviewer 1 Report

Abstract

Line 26: should there be a comma after challenge?

Line 28: should often and always be in italics to indicate it is a response from the survey?. Capital letter needed for While. 

Introduction

Lines 37-41: This is a rather long sentence - could this be restructured?

You mention play briefly and then discuss risky play - could you provide more discussion of the benefits of play? perhaps the different types of play? 

Line 80: Is this the first time you have used NZ? If so, write in full ?

Methods

Line 99: Do you need to state the online survey platform used?

Figure 1: good use of this figure - easy to follow. You excluded any parents with no children aged 5-12. Has this been made clear in the inclusion criteria? Without reading any further, I thought the survey was for any parent with a child under 18years? It may be my version but is the text a little 'fuzzy' in figure 1?

Is figure 1 repeated? On page 3 and 4 on my version?

Line 118 - You have put Sandseter et al (2007) - doesn't follow referencing format

Line 141: include reference for stuy validated survey for fathers?

Line 151: include reference for validation? Or is it reference 59?

Line 161: error message?

Results

Line 231 (and throughout): should it be 'parents' not 'people'?

Does table 3 need moving to page 8? 

Line 282: reword? i.e. '~The results show that as children age, the more likely...'

Line 290: error message?

Line 308: presumably used the bus? Can you say this? Was this stated in the suvery?

Table 4 : start on a new page so table not across two pages?

Discussion

Clear summary of results and discussion - interesting read, thank you. There may not be space, due to the wordcount but I wondered if it is worth exploring the impact on PA (maybe in terms of the type of risky play - thinking about fundamental movement skills) in more detail in your discussion as you mention PA in the introduction?  

Overall I feel this is a novel study and should contribute to the literature. I really enjoyed reading the paper, thank you.

Author Response

Reviewer 1

We thank the reviewer for their positive response to our paper and the recommendations they made to help us improve the quality of the manuscript. We have edited the stylistic comments, which we are grateful to the reviewer for pointing out, using track changes throughout the document. Where the reviewer makes comments about image quality, error messages and double prints of figures, we believe that this is a result of the submission process. Original figure files can be supplied to the journal for publication to ensure quality maintenance and links have been changed to plain text in order to avoid error messages.

In the introduction, we have expanded on the benefits of play in general (line 46-48) however, we have not gone into detail given the length of the paper. The reader is also referred to two excellent reviews in this field [1], [2]. We agree with the reviewer of the potential for risky play to benefit multiple aspects of physical activity however, we believe a discussion of these benefits would go beyond the scope of the paper which describes parents perceptions and practices of risky play. This is a valuable point that will be considered in our future paper which will explore factors that predict risky play and the likely benefits this has.

Kind regards,

Charlotte Jelleyman

[1] D. D. Whitebread, ‘The importance of play’, Toy Industries of Europe, p. 55, 2012.

[2] P. Gray, ‘The decline of play and the rise of psychopathology in children and adolescents.’, American Journal of Play, vol. 3, no. 4, pp. 443–463, 2011.

Reviewer 2 Report

I think my major concern is around confounding variables. Shouldn’t a regression analysis be conducted which is able to account for the demographic variables identified in table 2 which are likely confounders for the results? IF not should this have been planned? Is parent mental health or personality, the categories of risk and may be car ownership other important confounders? Is understanding levels of participation in sport also important to understand (e.g., taking part in rugby a lot may meet risky behavior need of child)? 

Until I can see this as a consideration I am not sure about the results. 

Introduction

I would give more time within the introduction to line 88-94 and describe this more and how i

Methods 

May be place design up front and say that you will report according to a guideline like STROBE

Separate out setting, eligibility criteria, procedures  

For Eligibility criteria

One dependent child less than 18 – co-variate is age.

Have a sub-title for sample size – identify the sample size calculation required – was this estimated?

2.2  state of play survey – not clear how this was used? Was it just used to define the different categories of risky play? Can you provide a reference for this survey so the questions can be examined 

the survey has reduced number of questionnaire for different age groups – what was the sample like and how was this represented check

analysis – just comments as above. 

Until I understand the above I have not gone further. 

Author Response

Reviewer 2

We thank the reviewer for his insightful comments regarding understanding participation in risky play. These are important questions that will be addressed in a future paper. However, given the large amount of data collected, the scope of the current manuscript was to describe perceptions, not to explain them. In order to make this clearer we have altered the title of the paper (A cross-sectional evaluation description of parental perceptions and practices related to risky play and independent mobility in children: The New Zealand State of Play Survey), stated more clearly our aims (line 94-100) and, as suggested by the reviewer, we have included a description of the study design (line 102-103). We hope that this explanation clarifies the purpose of the report. Furthermore, given that no statistical analyses were conducted on the data, we have not reported a required sample size, nor controlled for child’s age. These are factors that are being considered in the upcoming paper.

The risky play survey is a compilation of validated questionnaires (references have been provided where appropriate), questions commonly reported in independent mobility papers, and areas of interest of expert authors in this emerging field. The survey has now been added as an appendix to the paper (Additional File 2).

We have addressed the reviewer’s other comments in the following ways:

A STROBE checklist has been completed and, where necessary, the manuscript edited to follow the guidelines.

Section 2.1 Participants has been split into 3 sections; participants and setting and eligibility and inclusion criteria and procedures.

Kind regards,

Charlotte Jelleyman

Round  2

Reviewer 2 Report

Thank you for responding to the requests to update the paper. I have 

introduction

End of sentence line 115 – just check 

Methods

Much clearer thank you

Line 142 – add direction for reader i.e. please seeAdditional File 2…

Results 

Fine 

Discussion

Could limitations be added that relate to points made in the earlier review.  

Author Response

We are grateful to the reviewer for re-reading our paper and accepting our response to their comments. 

On our version of the manuscript, line 115 is not in the introduction, so we have re-read this section and made corrections where necessary. Furthermore, we have checked line 115 for errors.

Line 142 (now 127) has been changed to: "...independent mobility (please see Additional File 2 State of Play Questionnaire)."

The following sentence has been added to the discussion: " It would also be useful to understand what factors predict participation, and future research should attempt to identify which socio-demographic and individual characteristics influence risky play practices."

We hope that these corrections are satisfactory.

Kind regards,

Charlotte Jelleyman
